# Plant Selection for the Establishment of Push–Pull Strategies for *Zea mays–Spodoptera frugiperda* Pathosystem in Morelos, Mexico

**DOI:** 10.3390/insects11060349

**Published:** 2020-06-04

**Authors:** Ouorou Ganni Mariel Guera, Federico Castrejón-Ayala, Norma Robledo, Alfredo Jiménez-Pérez, Georgina Sánchez-Rivera

**Affiliations:** Laboratorio de Ecología Química de Insectos, Centro de Desarrollo de Productos Bióticos, Instituto Politécnico Nacional, Calle CeProBi No. 8, San Isidro, 62739 Yautepec, Mexico; nrobledo@ipn.mx (N.R.); aljimenez@ipn.mx (A.J.-P.); gsanchezri@ipn.mx (G.S.-R.)

**Keywords:** olfactometry, oviposition, feeding, survival, trap plant selection index (TRAPS)

## Abstract

Regulations imposed on the use of chemical insecticides call for the development of environmental-friendly pest management strategies. One of the most effective strategies is the push–pull system, which takes advantage of the behavioral response of the insect to the integration of repellent stimuli; it expels the pest out of the main crop (push), while attracting stimuli (attractants) pull the pest to an alternative crop or trap (pull). The objective of this study was to design a push–pull system to control *Spodoptera frugiperda* in maize crops (*Zea mays*) in Morelos, Mexico. Data on reproductive potential, larvae development, food consumption and olfactometry were used to obtain a Trap Plant Selection Index (TRAPS) based on Principal Component Analysis. This TRAPS was used to select the most suitable plants. The degree of repellency of potential plants to be used as the trap crop was studied with four-way olfactometers. *S. frugiperda* females oviposited more eggs on *Brachiaria hybrid* cv. Mulato II, *Panicum maximum* cv. Mombasa and *Panicum maximum* cv. Tanzania than on *Z. mays*, regardless of the fact that these plants delayed the development of their offspring. *Dysphania ambrosioides*, *Tagetes erecta* and *Crotalaria juncea* were less attractive to *S. frugiperda* females. Therefore, the former plants could be used as crop traps, and the latter as intercropped repellent plants in a push–pull system.

## 1. Introduction

Despite their limited neuronal capacity [1], herbivorous insects must face a continuous and challenging process of searching for, evaluating and selecting a host that may present mechanical and/or chemical defenses [2]. These defenses can be drastically affected by a domestication process that makes them generally more vulnerable than their wild relative [3,4,5]. Such is the case with maize, which is the result of a long process of domestication of teocintle (*Zea mays* ssp. parviglumis) begun approximately 10,000 years ago [6]. This process reduces their defenses [7], making them more susceptible to herbivores [8].

Native to the Americas and established in Europe [9], Africa [10,11] and Asia [12], *Spodoptera frugiperda* (Smith, J.E. 1797) (Lepidoptera: Noctuidae) is one of the generalist insects to have taken advantage of the low defenses of many domesticated crops (mainly grasses [13,14]) and become their key pest. Chemical control is the main way to combat these pests. However, the strict regulations imposed on chemical insecticides has increased the interest in environmentally friendly pest management strategies, among which are behavioral manipulation methods [15]. The latter consists of using chemical compounds that stimulate or inhibit the insect-pest’s behavior and, consequently, its expression [16]. Of these techniques, one of the few implemented that is effective is the push–pull strategy [17], which consists of manipulating the insect-pest’s behavior by using a repellent stimulus to expel it from the main crop, and an attractive stimulus to attract it to an alternative source, where it can be eliminated or controlled [18]. The successful design of an effective push–pull strategy requires knowledge of the insect-pest’s biology and its interaction with the candidate species to be integrated into the system. This knowledge should go beyond an understanding of repellent or attractive properties [19].

Plant colonization differs between specialist and generalist insects [20]. While specialists use more specific stimuli from a selected number of hosts, and are more tolerant of their host’s defenses [3,21,22], generalists such as *S. frugiperda* inhabit a wide range of hosts and are more sensitive to their defenses. Unlike specialists, generalists are less efficient in host selection and more vulnerable to their natural enemies [1]. Therefore, although generalists have advantages, such as greater availability of resources and a better nutritional balance [23], there is a general tendency in herbivorous insects to specialize [24]. Consequently, generalists are considered more vulnerable, more manipulable, and relatively easier to repel than specialists [3]. Fortunately, although most insects are specialists [1], most of those that become important agricultural pests are generalists. These present a host acceptability hierarchy, which is exploited in pest management strategy design, and mainly in trap plant selection [25].

In the establishment of push–pull systems, unlike the selection of repellent plants that is based mainly on antixenosis studies, attractive plant (trap) selection is based on a greater number of criteria (antixenosis, antibiosis and tolerance), generated by greater plant–insect interaction. Because of this larger number of criteria, trap plant selection can be highly subjective. Therefore, multivariate data reduction techniques can be used to construct selection indexes (SI) that simultaneously express these multiple dimensions in a simple way for interpretation [26]. Principal Component Analysis (PCA) is the best index construction method [27]. Based on their SI values, plants can be ordered according to characteristics that indicate their degrees of antixenosis (preference or non-preference), antibiosis and tolerance, in relation to a certain insect-pest.

The broad knowledge generated in recent decades regarding push–pull [28,29,30,31,32,33] and its successful use for the management of *S. frugiperda* in Africa [17,32,34], in addition to the empirical knowledge of Mesoamerican milpas (traditional local systems of maize polyculture), constitutes a foundation for the proposal of an efficient push–pull system for *S. frugiperda* management in Mexico. Therefore, this study selected attractive and repellent plants for the design of push–pull strategies for *S. frugiperda* management in maize crops (*Z. mays*), in the municipality of Yautepec, Morelos, México.

## 2. Materials and Methods

### 2.1. Insects

A *S. frugiperda* breeding stock of 117 larvae was obtained from maize crops at Yautepec (18°49′29.6″ N and 99°05′46.6″ W), Morelos, Mexico, in July 2018. These larvae were kept in plastic cups (3.5 cm in diameter and 3.8 cm high) in a bioclimatic chamber at the Insect Chemical Ecology lab, CeProBi-IPN, at 24 ± 2 °C temperature, 65% ± 5% relative humidity (RH) and a 12:12 light:darkness photoperiod. Larvae were fed an artificial diet [35]. After emergence, adults were kept in transparent acrylic boxes (20 × 20 × 20 cm) and fed with 10% honey solution in cotton pieces placed in 5 cm (diameter) circular containers at the bottom of the boxes.

### 2.2. Selection and Propagation of Plants in Greenhouse Conditions

We looked for plants that would fix nitrogen, had a commercial or nutritional value or could be used as green manure or forage, and were adaptable to Yautepec (altitude: 1210 MASL) and available in the area. In the case of potential repellent plants, we looked for host plants poorly colonized by this moth. The potential attractants (trap plants) selected were forage grasses of the genera *Brachiaria*, *Sorghum*, *Panicum* and *Lolium*, which are hosts of *S. frugiperda* [13,14] and widely used in the region [36,37]. The potential repellent plants considered were species of the genus *Tagetes*, *Dysphania* and *Crotalaria* [13,14]. Plants were propagated in greenhouses at 25 ± 2 °C, 65% ± 5% RH, using peat moss as the substrate in plastic pots (11 cm diameter, 9 cm height) and nursery trays (52 × 26 cm). Plants were watered as needed and after 15 days of emergence were fertilized with 8 g of Nitrofoska (N, 12%; P, 12%; K, 17%) by nursery tray. The characteristics of the plants (6 possible attractants and 4 possible repellents) are summarized in Table 1.

### 2.3. Oviposition Preference of Spodoptera frugiperda in Different Grasses

The oviposition preference, between maize and each of the potential attractant plants, was quantified in non-choice and two-choice experiments. Non-choice tests were carried out in 20 L × 20 W × 40 H cm transparent plexiglass arenas. A 30-day-old grass and two mated (24 h in advance) females were caged together. Arenas of the same material (90 L × 40 W × 40 H cm) were used for the two-choice tests. We released 4 mated (24 h in advance) females in the arena where two hosts (a maize plant and a grass) were placed on the floor 30 cm apart from each other. Females were allowed to oviposit for a scotophase (12 h) and the number of eggs oviposited per plant and per posture were quantified. Both tests were performed 10 times.

### 2.4. Influence of Trichome Density on Oviposition Preference

Grasses trichome density was assessed by a completely randomized design with a Species by Leaf Section factorial arrangement. The first factor had 7 levels (1. *Z. mays*; 2. *B. brizantha*; 3. *Brachiaria hybrid* cv. Mulato II; 4. *S. sudanense*; 5. *L. multiflorum*; 6. *P. maximum* cv. Tanzania; 7. *P. maximum* cv. Mombasa) and the second factor had 3 levels (1. Base; 2. Middle part; 3. Apex or tip) [38]. For each plant, the number of trichomes present in 1 cm^2^ of the abaxial surface of each section of the plant leaf was counted under a VE-S3 stereoscopic microscope. A total of 60 samples for each plant was counted.

### 2.5. Evaluation of Attraction and Repulsion of Spodoptera frugipeda Larvae to Plants

The insect response to volatile stimuli was evaluated in four scenarios (Figure 1). In scenarios I, II and III, two of the six possible attractants were evaluated, and in scenario IV, the four potential repellent plants were assessed simultaneously. Repellency was evaluated by combining each possible repellent plant with a proven attractant (*Z. mays*). The reduction of the attractive effect of maize to *S. frugiperda* larvae was indicative of the repellent effect of the associated plant.

The bioassays were performed at the beginning of the scotophase (red light, 60 watts; 60 Hz at 20 ± 3 °C and 65% ± 5% RH). A circular glass arena (diameter = 10 cm; height = 3 cm) with four arms was used in this experiment. Each airway received an airflow of 250 mL/min regulated by a flowmeter. The sources of volatile compounds (plants) were placed in glass bottles with two upper holes: one connected to the glass arena with a silicon tubing and the other equipped with an activated carbon filter, serving as an air-inlet hole (Appendix A).

To rule out directional bias presence in larval choice, prior tests were carried out with only controls (jars without plants). Therefore, before starting the bioassays, 20 third instar larvae were released in groups of six into the olfactometer arena, and their choices were recorded after 10 min [39]. Bioassays were carried out for several weeks. Each assay was carried out, releasing a single third instar larva in the center of the olfactometer arena. The choice of the larva was recorded in the 5 min that elapsed after the latency time. Only those larvae that had a 20 s minimum latency period and chose an odor source were registered. After every five replications, the olfactometer arena was rotated after being cleaned with cotton moistened with distilled water. We tested 80 third instar larvae for each scenario (16 assays per day) and took approximately 5 days (scotophase) to evaluate each scenario. The plants were taken from the greenhouse and installed minutes before the bioassays started. After the day’s assays, those plants were discarded.

### 2.6. Dispersion and Feeding Preference of Spodoptera frugiperda Larvae among the Different Grasses

These bioassays were performed at 25 °C ± 2 °C and 65% ± 5% RH. A total of 100 *S. frugiperda* neonates was released in the center of a circular arena (27 cm diameter; 10 cm height) containing 3 cm × 2 cm leaves, from each species to be tested, arranged circularly and equidistantly at the edges of the arena. The number of larvae on each leaf was counted after 6 h and the experiment was replicated five times.

The feeding preference of *S. frugiperda* larvae was studied through two-choice tests carried out by placing leaves of 3 cm^2^ (3 cm × 1 cm, obtained 30 min before from 30-day plants) in Petri dishes (9 cm diameter and 1.5 cm height) floored with a filter paper discs and a small cotton ball by the edge of the Petri dish (Figure 2). Filter paper discs and cotton balls were moistened with distilled water. The leaves were previously washed with water and dried at room temperature 15 min prior to experimentation. A first instar (L1) larva of *S. frugiperda* was placed in the center of each petri dish, equidistant to the leaves. We used 30 larvae in each of the six *Z. mays*–grass comparisons. Larval responses were recorded after 12 h (8 p.m. to 8 a.m.) and feeding preference was analyzed with the preference index (PI) used by Martínez et al. [40].

### 2.7. Performance of Spodoptera frugiperda on Potential Trap Plants

A breeding chamber at 24 ± 2 °C, 65% ± 5% RH, and with a 12:12 light:darkness photoperiod, housed all the Petri dishes (5.5 cm diameter by 1.5 cm height) used to measure the performance of *S. frugiperda* on the potential trap plants. The increase in larval weight in the grasses was evaluated by feeding 70 neonates of *S. frugiperda* (10 neonates/grass and 1 neonate per Petri dish) with leaves of each grass until prepupa stage. We also measured: (1) hatching period of *S. frugiperda* eggs measured in oviposited leaves collected in the oviposition bioassay (please see Section 2.3); (2) larva, prepupa, pupa and adult period length; (3) larval weight every 4 d until prepupa stage; (4) pupa weight 12 h after pupation; and (5) larval survival. This survival was studied in situ. We infested each of the 28 plants (5 neonates/plant and 4 plants/species) and recorded survival daily. Insects deaths were coded as one (1) and censured as zero (0).

### 2.8. Construction of Trap Plant Selection Index (TRAPS)

The average values of the number of oviposited eggs (∅), degree of attraction of larvae in olfactometer (ρ), feeding preference index (φ), larval growth performance (δ), larvae survival (υ) and cycle duration (ω) of *S. frugiperda* were used to construct a Trap Plant Selection Index (TRAPS) using the Principal Component Analysis (PCA) technique. The adequacy of the application of this technique was verified by Kaiser–Meyer–Olkin (KMO) test, the interpretation of which was based on Friel [41]. According to this author, a KMO between 0.90 and 1 is considered excellent; between 0.80 and 0.89 is good; between 0.70 and 0.79 is medium; between 0.60 and 0.69 is mediocre; between 0.50 and 0.59 is bad; and between 0 and 0.49 is totally inappropriate. A varimax orthogonal rotation was performed for a better interpretation of the components, and the number of retained main components was based on Kaiser criterion [42], which states that all extracted components must have an eigenvalue greater than or equal to one (1). These main components are linear combinations of the original variables, and the construction of the TRAPS index was based on the first principal component defined as:(1)$$TRAPS=\alpha_{11}*\emptyset +\alpha_{12}*\rho +\alpha_{13}*\varphi +\alpha_{14}*\delta +\alpha_{15}*\upsilon +\alpha_{16}*\omega$$
in which TRAPS is the Trap Plant Selection Index (the larger its values, the more suitable the plant will be as a trap plant) and $a_{1i}$ are the autovector elements that represent the contributions (weights) of each variable to the first principal component.

### 2.9. Statistical Data Analysis

In the non-choice oviposition tests, the number of eggs/plant and the number of eggs/posture were analyzed with an ANOVA test (Tukey, α = 0.05). In the two-choice tests, the number of oviposited eggs per plant was analyzed with a student’s *t*-test (α = 0.05) and an Oviposition Preference Index (OPI), proposed by Fenemore [43] and calculated using expression 2. The OPI ranges between +100 (fully stimulating), 0 (no effect) and −100 (fully dissuasive).
(2)OPI=[(A−B)/(A+B)]∗100
in which A is the number of oviposited eggs on the species evaluated and B is the number of oviposited eggs on the reference species (control).

Trichome density was analyzed by a 2-way ANOVA test (Tukey, α = 0.05) and the association between it and oviposition preference was analyzed by the point biserial correlation coefficient [44]. The attraction or repulsion of *S. frugiperda* larvae to the evaluated plants was analyzed by the Friedman test, and the pairwise comparisons of means were by the Wilcoxon signed-rank test.

In the larvae dispersion bioassay, the numbers of *S. frugiperda* neonates on the grasses were analyzed with an ANOVA test (Tukey, α = 0.05) and the inequality in their distribution was assessed with Lorenz [45] curve and Gini [46] concentration index (I_G_) (Expression 3), which results were interpreted based on the scale established by Noce et al. [47].
(3)$$I_{G}=\overset{n-1}{\underset{i=1}{\sum }}(P_{i}-Q_{i})/\overset{n-1}{\underset{i=1}{\sum }}P_{i}$$
in which:$\;P_{i}=$ cumulative proportion of leaves; $Q_{i}=$ cumulative proportion of neonates on the leaves; n = number of species (leaves); $X_{i}$ = leaf area of each species i.

In the feeding preference bioassays, the indices (PI) obtained were subjected to angular transformation ($Arcsen\sqrt{PI}$) and analyzed by one sample *t*-test, testing as a null hypothesis $H_{0}:\mu \;\left(PI\right)=Arcsen\sqrt{0.5}=0.79$ (there is no preference between the control and the species evaluated) and as an alternative hypothesis $H_{a}:\mu \;\left(PI\right)\neq 0.79$. When rejecting the null hypothesis, PI < 0.5 indicates preference for control, and PI > 0.5 indicates preference for the species evaluated [40].

Increase in larval weight was analyzed by adjusting sigmoidal growth models to obtain the best equation for each grass. Gompertz [48], Richards [49] and Verhulst (Logistic) [50] nonlinear regression models (Appendix A) were adjusted using iterative methods of Levenberg–Marquardt or Gauss–Newton according to their convergence. The goodness of fit of the obtained equations was evaluated using the adjusted coefficient of determination ($R_{aj}^{2}$), the Root of the Mean Square Error (RMSE) and the compliance with the assumptions of normality, homocedasticity and independence of residues, verified by the tests of Kolmogorov–Smirnov [51,52], White [53] and Durbin–Watson [54], respectively.

All the variables related to the development cycle (durations of the development stages and pupae weight) were analyzed by ANOVA tests (Tukey, α = 0.05). Larvae survival in each grass was analyzed by the Kaplan–Meier method [55]. Survival curves were compared by Wilcoxon test and pairwise comparisons by Holm–Sidak test (α = 0.05).

All parametric tests were preceded by the verification of compliance with the assumption of normality by the Kolmogorov–Smirnov test [51,52] and the assumption of homoscedasticity by the Levene’s test [56]. All analyses and regression model adjustments were carried out on SPSS v. 20 [57].

## 3. Results

### 3.1. Oviposition Preference of Spodoptera frugiperda to Different Grasses

The non-choice bioassays showed a significant difference between the number of eggs oviposited (F = 13,632; df = 6; *p* < 0.01) among the plants, these being higher for *P. maximum* cv. Mombasa. The second group was formed of *Z. mays* and *S. sudanense*. Females laid similar numbers of eggs on *B. hybrid* cv. Mulato II, *P. maximum* cv. Tanzania and *L. multiflorum*. The least number of eggs was oviposited on *B. brizantha* (Figure 3A). Regarding the number of eggs per posture, a significant difference was observed too (F = 10.702; df = 6; *p* < 0.01). The values were similar in most grasses, varying between 146.93 ± 13.89 in *P. maximum* cv. Mombasa and 107.70 ± 7.73 in *B. hybrid* cv. Mulato II. The least number of eggs per posture was observed on *B. brizantha*. (Figure 3B). In the two-choice tests, the oviposition preference index values were 5.09, 38.13, 4.94, 31.2, 26.72 and −25.34 for the species *Tanzania*, *Mombasa*, *Mulato II*, *L. multiflorum*, *S. sudanense* and *B. brizantha*, respectively. In the presence of the visual, chemical and tactile stimuli provided by the plants, *S. frugiperda* preferred to oviposite on *P. maximum* cv. Mombasa (*t* = −2.481; df = 18; *p* = 0.023), *L. multiflorum* (*t* = −2.591; df = 18; *p* = 0.018) and *S. sudanense* (*t* = −3.486; df = 18; *p* = 0.003) than on *Z. mays*, but preferred the latter over *B. brizantha* (*t* = −3.014; df = 18; *p* = 0.007) (Figure 4).

It was observed that *S. frugiperda* preferred to oviposite on the leaves’ abaxial surface. Only 4.07% of the postures were found on the adaxial surfaces of the species, except for *Z. mays* and *B. brizantha*, which received all their postures on their abaxial surfaces. 32.82% of the postures were placed on the arena surfaces.

### 3.2. Influence of Trichome Density on Oviposition Preference

Trichome density varied according to species (F = 246.521; df = 6; *p* < 0.0001), leaf section (F = 12.975; df = 2; *p* < 0.0001) and its interaction (F = 6541; df = 12; *p* < 0.0001). This interaction (species * section) was analyzed, and the results summarized in Appendix A. In all sections, Mulato II (Group I) and *B. brizantha* (Group II) trichome densities were higher than those of the other species. Except *P. maximum* cv. Mombasa, in all species the leaf bases trichome densities were higher than those of the other sections. *S. sudanense*, *L. multiflorum* and *Z. mays* presented similar trichome densities in all sections.

The degrees of association between oviposition preference and trichome density are presented in Table 2. The point biserial correlation coefficient was only calculated for tests in which there was oviposition preference.

### 3.3. Response of Attraction or Repulsion of Spodoptera frugiperda Larvae

In the test with no plants, 71.67% of the larvae remained in the center of the arena, and the remaining 28.33% were randomly distributed among the four olfactometer pathways (p=0.824; $\chi^{2}=0.905;n=20;\mathrm{df}=3$). The results of the bioassays in the different scenarios are presented in Figure 5.

In each scenario, the larvae responded differently to volatile sources (Scenario I—p=0.037; $\chi^{2}=8.5;n=80;\mathrm{df}=3$; Scenario II—p=0.000147; $\chi^{2}=20.30;n=80;\mathrm{df}=3$; Scenario III—p=0.011; $\chi^{2}=11.10;n=80;\mathrm{df}=3$). In all these scenarios, the numbers of larvae attracted to negative controls were significantly lower than those attracted by grasses. In scenarios I and III, the larvae’s responses to grass volatiles were similar to those to maize. In scenario II larvae preferred *P. maximum* cv. Mombasa (Z = −2.359; *p* = 0.018) and *P. maximum* cv. Tanzania (Z = −1.982; *p* = 0.047) over maize. In scenario IV, a significant difference was observed between the maize attraction reductions enacted by the candidates for repellent plants (p=0.001; $\chi^{2}=16.059;n=80;\mathrm{df}=3$). The greatest reduction was produced by *D. ambrosioides*, the repellent effect of which did not differ from those of *C. juncea* and *T. erecta*. *T. lucida* was less repellent, allowing the attraction of 25 larvae by the maize plant.

### 3.4. Dispersion of Spodoptera frugiperda Larvae on Grasses

A Gini index of 0.318 was considered average according to Noce et al. [47], and the distance of the Lorenz curve from the equidistribution line indicates the concentration of neonates in some of the grasses (Figure 6A). Most of the neonates concentrated on maize (27.06%), *S. sudanense* (23.53%) and *L. multiflorum* (16.47%) (Figure 6B).

### 3.5. Feeding Preference of Spodoptera frugiperda Larvae on the Grasses

The *S. frugiperda* feeding preference tests results are presented in Figure 7. The results indicated that the larvae had a high preference for *L. multiflorum* and *S. sudanense*, and less preference for *P. maximum* cv. Mombasa, *P. maximum* cv. Tanzania, *B. brizantha* and *B. hybrid* cv. Mulato II presented preference indices like those of maize.

### 3.6. Performance of Spodoptera frugiperda in Potential Trap Plants

The estimates of the parameters for the adjusted models for the prediction of larval weight increase in each one of the grasses are in Table 3. All the models presented RMSE less than 3% and an $R_{\mathrm{aj}}^{2}$ greater than 90%. The logistic model was better adjusted to the larval weight in maize, *P. maximum* cv. Mombasa, *S. sudanense* and *P. maximum* cv. Tanzania. In *L. multiflorum* and Mulato II, the data generated adhered better to the Gompertz model, and those obtained from *B. brizantha* to Richards model. All the equations complied with the assumptions of normality, homocedasticity and independence of residues, which indicates that the results of the F and *t*-tests, applied to verify the significance of the equations and their respective parameters, are consistent. These equations (Table 3) generated larval weight increase curves, and declared at day 15 (larval stage duration on maize) three levels of growth: a high level provided by maize, a medium level provided by *L. multiflorum* and *S. sudanense*, and a low one provided by the rest of the species (Figure 8A). The larval stage was shorter in *Z. mays*, *L. multiflorum* and *S. sudanense*. Maize produced the heaviest larvae and is also the species that gave the larvae the largest increase in weight whose highest rate would be reached at 17 days (Inflection point, Figure 8B).

The durations of the different development stages of *S. frugiperda* in each of the grasses are presented in Table 4. The stages of egg and larva had longer durations in *B. brizantha*, *Mombasa*, *Tanzania* and Mulato II. The durations of the prepupa, pupa and adult stages did not differ between species. Insects fed on *Z. mays* and *L. multiflorum* had a significantly shorter life cycle than the others, but the former produced the heaviest pupae and *B. brizantha* the lightest ones.

The probabilities of survival of the neonates were higher in *Z. mays*, *S. sudanense* and *L. multiflorum*, respectively (Wilcoxon test; $\chi^{2}=55.57;\mathrm{df}=6;p<0.0001$) (Figure 9), where the first two had survival times over 7 days. *P. maximum* cv. Tanzania, *B. brizantha*, *B. hybrid* cv. Mulato II and *P. maximum* cv. Mombasa presented survival times below 5 days, and survivals rates below 0.25.

### 3.7. Trap Plant Selection Index (TRAPS)

The KMO value of 0.788, considered median according to the Friel [41] classification, and Bartlett’s sphericity test ($\chi^{2}=32.529;\mathrm{df}=15;p=0.005$) indicate that Principal Component Analysis is applicable to the data. Based on the Kaiser criterion (eigenvalue greater than one), two principal components (PC1 and PC2) were extracted, which explain 79.71% of the original variables’ variance (Table 5). The principal components matrix is presented in Table 5.

A suitable trap plant is one that is preferred by mated females to oviposit, but is not preferred by larvae, and does not favor their growth. TRAPS construction was based on the first principal component according to Table 5. The coefficients associated with feeding, survival and cycle duration of the larvae, in the second principal component, were inconsistent. The negative sign of a coefficient associated with a variable indicates that it represents an undesirable characteristic, which is not the case for this variable (cycle duration), the higher values of which are preferable. The positive signs of the variables feeding and survival, the higher values of which are undesirable, also invalidate PC2. Therefore, the following TRAPS index was proposed based on PC1, the coefficients of which are consistent.
(4)TRAPS=0.829∗∅+0.723∗ρ − 0.545∗φ − 0.932∗δ − 0.578∗υ+0.919∗ω

This index allowed grass classification, from the most to the least suitable as a trap plant for *S. frugiperda* (Table 6). *P. maximum* cv. Mombasa, *P. maximum* cv. Tanzania and *Brachiaria hybrid* cv. Mulato II are the most suitable species as trap plants of *S. frugiperda*, as they are preferred by females to oviposit, but they are not very suitable for the development of their offspring.

The combination of the trap plants (*B. hybrid* cv. Mulato II, *P. maximum* cv. Mombasa and *P. maximum* cv. Tanzania) with the three species repellent to *S. frugiperda* (*D. ambrosioides*, *T. erecta* and *C. juncea*) generated nine push–pull strategies (Figure 10).

## 4. Discussion

Plant selection for the design of push–pull strategies was carried out by studying the relationship of *S. frugiperda* with potential attractive or repellent plants. *B. brizantha* was the only grass with an oviposition deterrent effect (indicated by negative OPI index), the other grasses stimulated oviposition (Figure 3 and Figure 4). The oviposition preference for *P. maximum* cv. Mombasa, *L. multiflorum* and *S. sudanense* agrees with Pitre et al. [58], who report *L. multiflorum* among the grasses preferred by *S. frugiperda* for oviposition. This species preferred to oviposit on the leaves’ abaxial surfaces, as reported by other authors (Pitre et al. [58], Ali et al. [59] and Beserra et al. [60]).

Trichome density is one of the main causes of antixenosis in generalist lepidoptera oviposition. In this study, the highest densities were observed in *B. hybrid* cv. Mulato II and *B. brizantha* (Appendix A). The values found for these species are lower than those reported by Cheruiyot et al. [61]. Except for Mombasa, the trichome density decreased from the leaf base to the leaf apex, as reported by Rendón-Carmona et al. [38].

The *B. brizantha–Z. mays* assay revealed a strong negative correlation between trichome density and oviposition (Table 2). Preference for the species with the lowest density of trichomes may indicate that these structures inhibit oviposition, as reported by Kumar [62]. However, this result differs from those of Pitre et al. [58], who associate *S. frugiperda* oviposition preference with leaf color. Negative correlations were also observed in the *P. maximum* cv. Mombasa and *S. sudanense* assays. On the other hand, a significant positive correlation was observed between oviposition and trichomes density in the *L. multiflorum–Z. mays* test, which could indicate that females preferred to oviposit in plants with higher trichomes density, or that trichomes were not determinants for oviposition, as the densities observed in both species were low. It was also perceived that the high density of trichomes of *B. hybrid* cv. Mulato II could not inhibit oviposition, which was like the density observed in maize. Volatile compounds attractive to *S. frugiperda* and released by Mulato II may have counteracted the supposed trichome inhibitory effect. This leads us to infer that oviposition site selection may depend on the balance of physical or tactile stimuli, chemical stimuli (volatile compounds) and visual stimuli (color and shape) [20]. Kumar [62] asserts that tactile stimuli are more decisive. For *S. frugiperda*, Rojas et al.’s [63] results confirm the greater importance of these tactile stimuli (rough surfaces) over chemicals (leaf volatiles). These authors report that *S. frugiperda* prefers to oviposit on rough surfaces, not to be confused with trichomes, which are plants’ defense structures. From the inconclusive results of the assays with Mulato II and *L. multiflorum*, it follows that, although tactile stimuli (in this case the trichomes) are decisive, they can be weighted by the females in the presence of chemical attractants. Among the grasses evaluated, *S. frugiperda* preferred *P. maximum* cv. Mombasa, *L. multiflorum* and *S. sudanense* over maize. Although the amounts of oviposited eggs in Tanzania and Mulato II were not significantly higher than those recorded in maize, they were higher, and by this criterion (oviposition), it is considered that they would be optimal *S. frugiperda* trap plants, as well as *P. maximum* cv. Mombasa, *L. multiflorum* and *S. sudanense*. *B. brizantha* would be inadequate as a trap plant in the push–pull design for the pathosystem *Z. mays–S. frugiperda*.

The oviposition site chosen by *S. frugiperda* may be inappropriate for larval development [64], as the neonates can leave that oviposition site by walking or ballooning [65]. Neonates, despite their limited olfactory system, use chemical signals that are crucial for host selection, mainly in their early stages [66]. In the olfactometry studies carried out in this study, the attraction exerted by Mombasa and Tanzania on *S. frugiperda* larvae (Figure 5) is probably due to green leaf volatiles (GLVs) that are generally the mediators of generalist Lepidoptera attraction to Poaceae [67]. Among these GLVs stand out α-pinene and linalool, which have proven to be generators of significant antennal responses in *S. frugiperda* [68]. A similar trend was observed with *S. sudanense* and Mulato II, although the attractions of these were not significantly superior to those of maize.

The repellent activity of *T. erecta* (Figure 5) agrees with Díaz and Serrato [69], who state that species of the genus *Tagetes* contain, in their aerial parts, secondary metabolites that can be repellent and/or toxic for numerous insect-pests. Several studies report repellent activities of *T. erecta* against many lepidopteran pests. Among them are those of Calumpang and Ohsawa [70], who found that this species is repellent to *Leucinodes orbonalis*, and attributed that repellency to citral and 1-dodecene, two of the seven volatile organic compounds found in the GC–MS (Gas Chromatography–Mass Spectrometry) analysis of that species. In addition to the repellent activity found here, *T. erecta* also has an antialimentary activity that can cause up to 72% mortality of *S. frugiperda* larvae [71]. This capacity, added to its great utility, makes this species a crop with great potential for pest management through behavioral manipulation strategies such as push–pull. The repellency of *C. juncea* to *S. frugiperda* can be attributed to the presence, in its leaves, of triterpenes, alkaloids, flavonoids and phenolic compounds, as reported by Al-Snafi [72]. The repellent activity of *C. ambrosioides* is probably due to the volatile terpenoid compounds reported by Sagrero-Nieves and Bartley [73], which are generally repellents to herbivorous insect. For a better understanding of the relationship of *S. frugiperda* with the plants analyzed here, more detailed studies (GC–MS analysis and electrophysiological bioassays) will be necessary. In the larval dispersion bioassay, the uneven distribution of neonates observed (Figure 6) may be indicative of the presence of toxins in the grasses [65].

The lower growth rate in larval weight in the other grasses compared to maize resulted in an extension of the duration of larval stages (Figure 8; Table 3 and Table 4). This can be attributed to the low nutritional quality of the plant tissues of these species and/or a higher density of trichomes, mainly in Mulato II and *B. brizantha*, which may have inhibited the feeding of *S. frugiperda* larvae [74]. Secondary metabolites can also influence larval performance by being stimulants or deterrents of their feeding. This is probably the case of Mombasa, the feeding preference of which was lower than that of maize (Figure 7) despite its low trichomes density. Low host quality is compensated by the lengthening of larval stage duration [75,76], as occurred with Mombasa, Tanzania, Mulato II and *B. brizantha*. The increase in the length of the life cycle (Table 4) would reduce the population growth of the insect-pest and increase its exposure to natural enemies [77]. On the other hand, the larval stage of *S. frugiperda* was shorter in *Z. mays*, *L. multiflorum* and *S. sudanense*, due to a higher rate of larval development in these species. This latter will produce a greater number of generations per year, which would favor the population growth of the pest.

According to Slansky and Feeny [78], the low nutritional quality mentioned above could lead to higher larval mortality, as in our study. On the fifth day of the survival assessment, the species Mombasa, Tanzania, Mulato II and *B. brizantha* produced larvae mortality rates of 90%, 75%, 85% and 55%, respectively (Figure 9). This high mortality of neonates and larvae of the first two instars is probably due to the antialimentary, insecticidal, antimicrobial and allelopathic [79,80,81] effect of benzoxazinoids [82,83]. *S. frugiperda* larvae ingesting these phytochemicals (DIMBOA and MBOA) detoxify them via N-glycosylation [84], preventing them from having negative effects on their growth [85]. However, in some grasses, the combined effect of chemical (including benzoxazinoids at different concentrations) and physical (trichome) defenses can affect neonates’ growth, and even become lethal. Numerous factors can affect the concentrations of DIMBOA in grasses, age being one of them. Generally, higher concentrations of these compounds are found in younger plants [82,86], information that must be considered for the timing of the establishment of grasses as trap plants in push–pull systems in the field. The high larval mortality before reaching the most voracious larval stages (third and fourth instars for *S. frugiperda*) (Figure 9), and the lengthening of the larval period (greater exposure to natural enemies) of those who survived (Table 4), could inhibit pest development in the grasses [87].

*L. multiflorum*, *B. brizantha* and *S. sudanense* fulfilled the hypothesis of “the mother knows best”, according to which the females oviposit in plants suitable for the development of their offspring [88,89]. However, this hypothesis, also known as Jaenike’s preference–performance hypothesis, is not always met, and the larvae of some species can make their own choices if those of their parents are not suitable for their development [66,90]. Females can prioritize their needs over those of their offspring in oviposition site selection, and this is known as the “optimal bad motherhood” principle [91,92,93]. It was observed that Mombasa, Tanzania and Mulato II were preferred by the females, but these grasses provided poor survival, and poor and slow development, to their offspring. In generalist insects such as *S. frugiperda*, this can occur due to its wide range of potential hosts [94], oviposition pressure [95], the mobility of the larvae from one plant to another to correct the poor choice of their parent [96], Hopkins’ host-selection principle [97], and the enemy-free space hypothesis [98], which leads some species to oviposit outside their host plant, possibly as a strategy to evade their natural enemies. It may be due to the latter hypothesis that *S. frugiperda* tends to oviposit outside its host plants, both in experimental conditions (observations in this study and results of Rojas et al. [63]) and in the field.

The nine push–pull strategies (Figure 10) can be used for *S. frugiperda* management in maize crops in Morelos, and probably in other regions where the plants that make up these systems develop properly. However, its field effectiveness needs to be studied in future studies.

## 5. Conclusions

The TRAPS index based on the Principal Component Analysis technique allowed the classification of grasses, in decreasing order of suitability as trap plants in push–pull systems.

In addition to their high economic value as forages, the grasses *B. hybrid* cv. Mulato II, *P. maximum* cv. Mombasa and *P. maximum* cv. Tanzania are good candidates for use as trap crops in push–pull systems because they were preferred by *S. frugiperda* females for ovipositing, and they did not offer good nutritional or survival conditions for their offspring.

The culinary value of *D. ambrosioides*, ornamental and religious values of *T. erecta*, and the value of *C. juncea* as a nitrogen fixer, forage and green manure, in association with their ability to make maize less attractive to *S. frugiperda*, make these species suitable for intercropping in push–pull systems.

The combination of these species generated nine push–pull strategies, the field effectiveness of which against *S. frugiperda* in maize crops needs to be evaluated in future studies.

## Figures and Tables

**Figure 1 insects-11-00349-f001:**
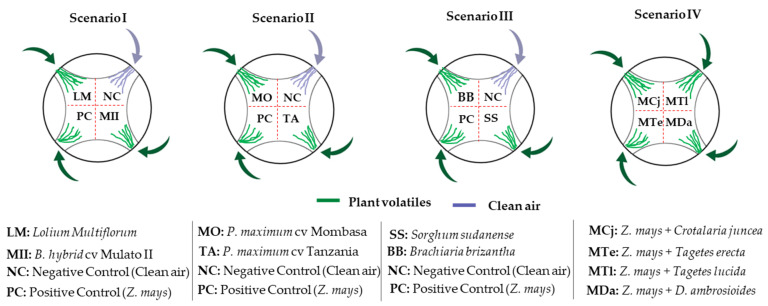
Scenarios considered in attraction (scenarios I, II and III) and repulsion (scenario IV) bioassays of third instar (L3) larvae of *S. frugiperda* in the presence of potential attractant or repellent plants in a 4-way olfactometer.

**Figure 2 insects-11-00349-f002:**
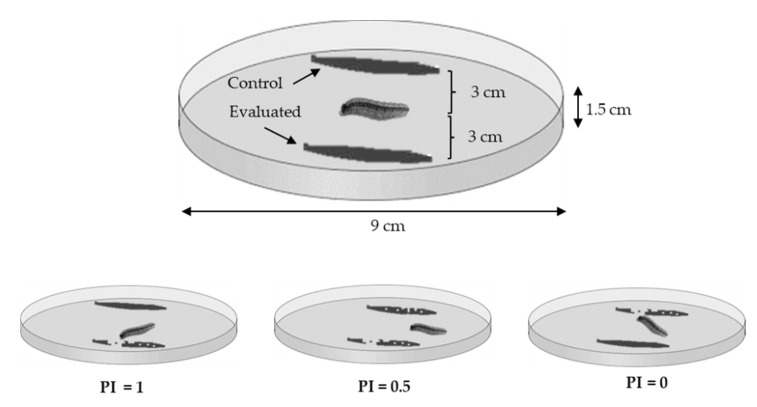
Device for feeding preference assessment of first instar larvae of *S. frugiperda* (not at scale).

**Figure 3 insects-11-00349-f003:**
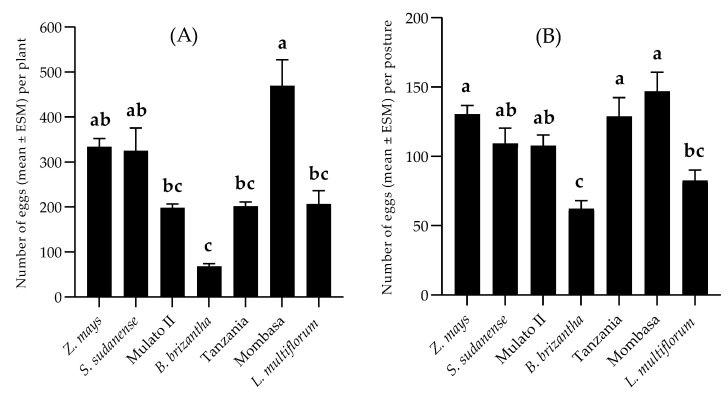
Number of eggs (Mean ± SEM) oviposited per plant (**A**) and per posture (**B**) by *S. frugiperda* in no-choice test. Different letters atop the bars indicate significant differences (Tukey test, α = 0.05).

**Figure 4 insects-11-00349-f004:**
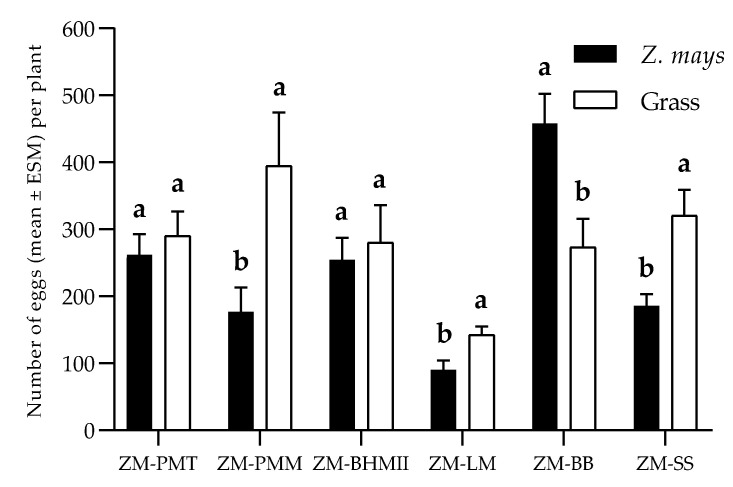
Number of eggs (Mean ± SEM) oviposited per plant by *S. frugiperda* in two-choice test. Different letters indicate the existence of significant difference in the *t*-test (*p* < 0.05). ZM = *Z. mays*; PMT = *P. maximum* cv. Tanzania; PMM = *P. maximum* cv. Mombasa; BHMII = *B. hybrid* cv. Mulato II; LM = *L. multiflorum*; BB = *B. brizantha* and SS = *S. sudanense*.

**Figure 5 insects-11-00349-f005:**
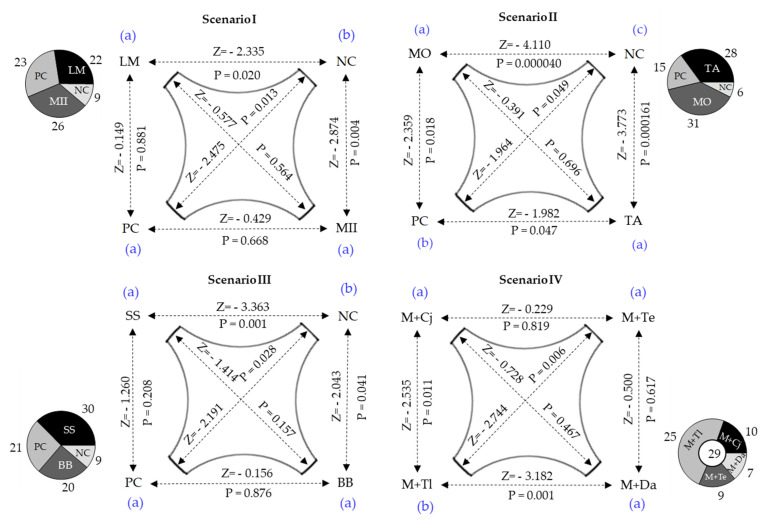
Behavioral responses of attraction (Scenarios I, II and III) and repulsion (scenario IV) of *S. frugiperda* larvae (L3) in presence of potential attractive [LM = *L. multiflorum*, MII = *B. hybrid* cv. Mulato II, MO = *P. maximum* cv. Mombasa, TA = *P. maximum* cv. Tanzania, SS = *S. sudanense*, BB = *B. brizantha*, NC = Negative Control, PC = Positive Control (*Z. mays*)] or repellent plants [M + C_j_ = *Z. mays* + *C. juncea*; M + Te = *Z. mays* + *T. erecta*; M + Tl = *Z. mays* + *T. lucida*; M + Da = *Z. mays* + *D. ambrosioides*]. Z = Wilcoxon signed-rank test preceded by Friedman test (α = 0.05). For each scenario, different letters at the corners indicate significant difference (Wilcoxon signed-rank test, α = 0.05). Pie graphs besides each scenario represent the number of larvae responding to each condition. The circle in the middle of the scenario IV pie graph represents the number of larvae that made no choice.

**Figure 6 insects-11-00349-f006:**
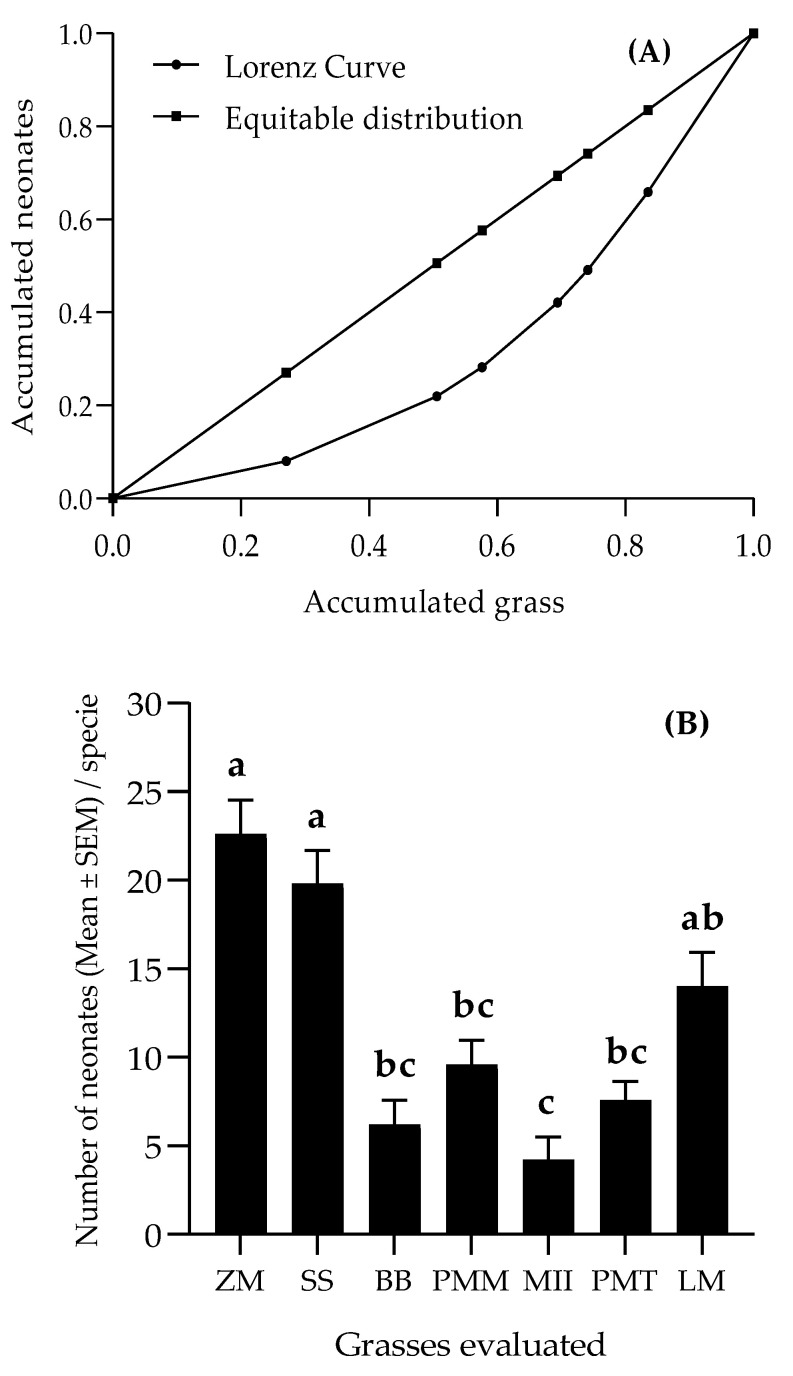
Lorenz curve (**A**) and histograms (**B**) of distribution of *S. frugiperda* neonates in the grasses. Bars topped by different letters indicate significant difference by the Tukey test (α = 0.05). (ZM = *Z. mays*; SS = *S. sudanense*; BB = *B. brizantha;* PMM = *P. maximum* cv. Mombasa; MII = *B. hybrid* cv. Mulato II; PMT = *P. maximum* cv. Tanzania; LM = *L. multiflorum*).

**Figure 7 insects-11-00349-f007:**
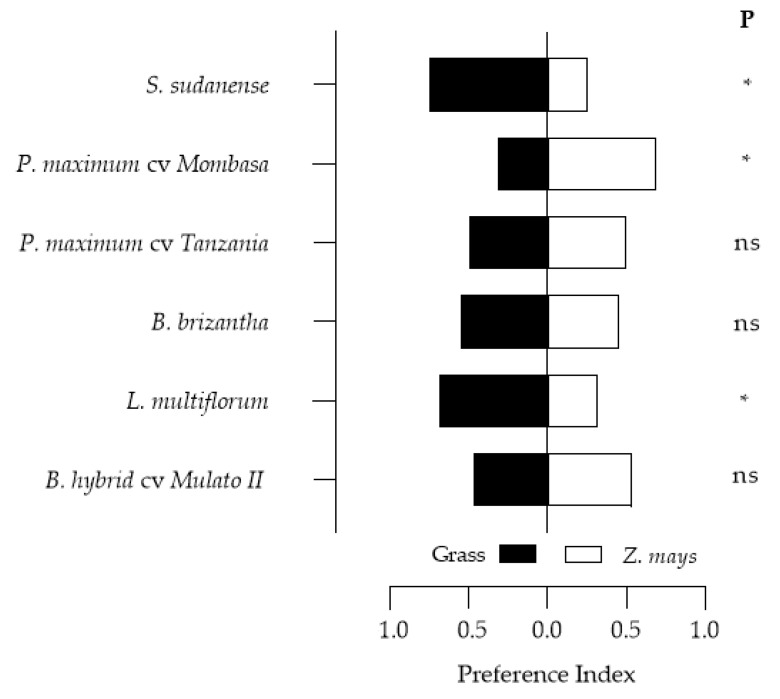
Feeding preference indices of *S. frugiperda* larvae in two-choice tests. * *p* < 0.05 (significant difference), ns = no significant difference.

**Figure 8 insects-11-00349-f008:**
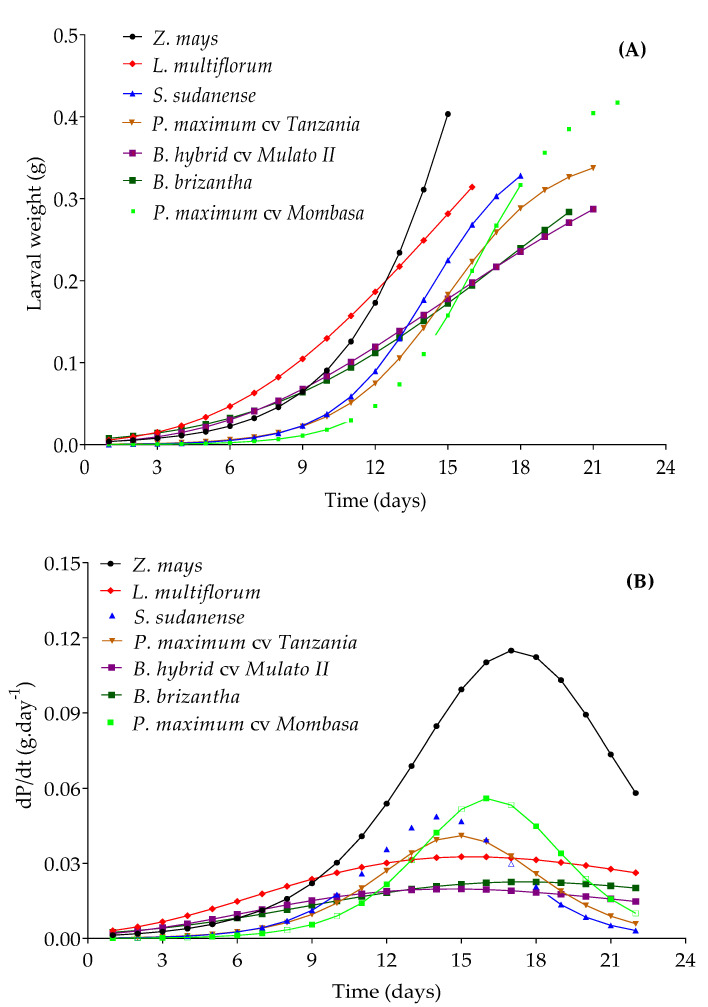
Growth curve (**A**) and growth rate (**B**) for *S. frugiperda* larvae weight in different grasses. Curves were compared at day 15. dP/dt = derivative of the weight growth function with respect to time, which represents larval weight growth rate.

**Figure 9 insects-11-00349-f009:**
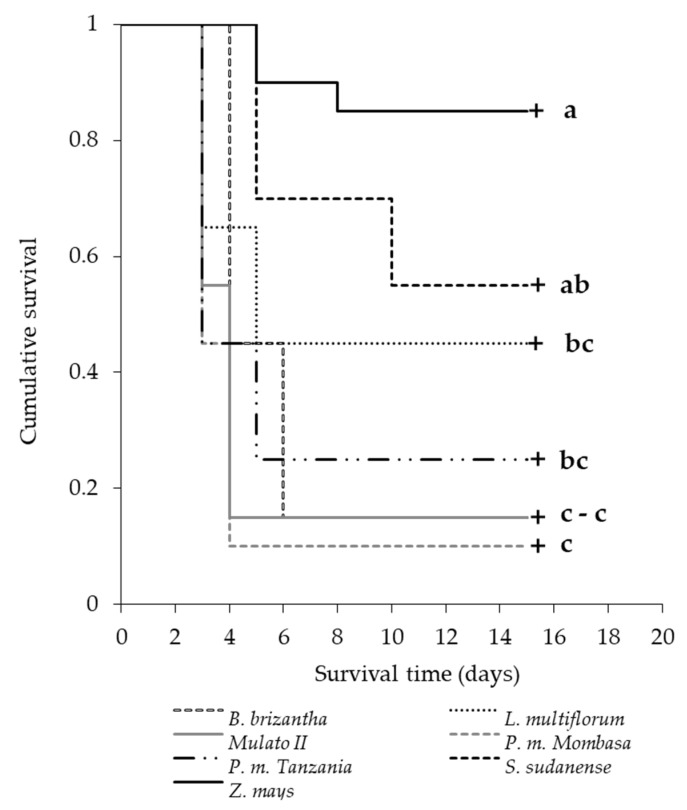
Survival of *S. frugiperda* larvae in different grasses. Lines with different letters indicate significant differences by the Holm–Sidak test (α = 0.05).

**Figure 10 insects-11-00349-f010:**
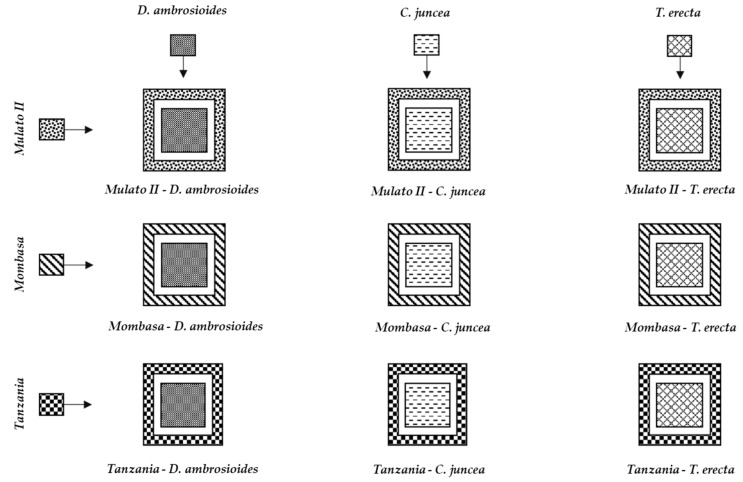
Proposed push–pull strategies for fall armyworm management in maize crops in Yautepec, Mor., Mexico.

**Table 1 insects-11-00349-t001:** Characteristics of potential attractant or repellent plants, tested for the design of push–pull strategies to manage *S. frugiperda* at Yautepec, Morelos, Mexico.

	Plants	Origin	Plant Height (m)	Altitude (m)	Main Uses
**Attractants**	*Brachiaria brizantha* ^‡,§^	Uganda, Botswana	2.50	0–1800	Forage, Live Barrier
	*Panicum maximum* cv. Tanzania ^‡,^*	Tropical and subtropical Africa	2.00	0–1800	Forage, grazing
	*Panicum maximum* cv. Mombasa ^‡^,*		2.50	0–2000	
	*Brachiaria hybrid* cv. Mulato II ^‡, §^	3 generations of brachiaria pasture crossing	1.00	0–1800	Forage
	*Lolium multiflorum*^‡,†^,*	Africa, Europe and Asia	1.30	0–2400	Forage, Lawns
	*Sorghum sudanense* ^‡,§^	Africa	3.00	0–1800	Forage, Cut
**Repellents**	*Dysphania ambrosioides **	America	1.00	0–3000	Condiments, Medicinal
	*Crotalaria juncea **	India and Pakistan	2.40	0–1900	Forage, green manure
	*Tagetes erecta **	Mesoamerica	1.00	800–2300	Ornamental, Religious, Medicinal
	*Tagetes lucida **	Mexico and Guatemala	0.80	800–2700	

^‡^ [13]; ^†^ [14]; *preselected in this study to be integrated in the push–pull system; ^§^ already proposed in other push–pull studies.

**Table 2 insects-11-00349-t002:** Trichome density influence on oviposition preference of *S. frugiperda*.

Species Evaluated/Reference	Preference	$r_{pb}$	*t*	*p*
*P. maximum* cv. Tanzania/*Z. mays*	No	-	-	-
*P. maximum* cv. Mombasa */*Z. mays*	Yes	−0.156	−1.712	0.089
*B. hybrid* cv. Mulato II/*Z. mays*	No	-	-	-
*L. multiflorum* */*Z. mays*	Yes	0.441	5.338	<0.0001
*B. brizantha/Z. mays* *	Yes	−0.765	−12.884	<0.0001
*S. sudanense* */*Z. mays*	Yes	−0.094	−1.021	0.310

* Preferred species; $r_{pb}$ = point biserial correlation coefficient between preference and trichome density.

**Table 3 insects-11-00349-t003:** *S. frugiperda* larval weight growth equations on different grasses.

Species	Selected Model	Parameter Estimates				Inflection Point	$R_{aj}^{2}$ (%)	RMSE (%)	*p*
		${\hat{\beta }}_{0}$	${\hat{\beta }}_{1}$	${\hat{\beta }}_{2}$	${\hat{\beta }}_{3}$				
*Z. mays*	Logistic	1.277 *	480.94 *	0.360 *	-	*t* * = 17.15 days	95.00	2.28	<0.05
						P*_t_* = 0.64 g			
*P. maximum* cv. Mombasa	Logistic	0.438 *	3839 *	0.512 *	-	*T** = 16.12 days	98.48	1.71	<0.05
						P*_t_* = 0.22 g			
*S. sudanense*	Logistic	0.373 *	1705 *	0.524 *	-	*t* * = 14.20 days	98.85	1.23	<0.05
						P*_t_* = 0.18 g			
*P. maximum* cv. Tanzania	Logistic	0.358 *	932.6 *	0.459 *	-	*t* * = 14.91 days	98.84	1.35	<0.05
						P*_t_* = 0.18 g			
*L. multiflorum*	Gompertz	0.796 *	1.712 *	0.112 *	-	*t* * = 15.34 days	94.82	2.42	<0.05
						P*_t_* = 0.29 g			
*B. hybrid* cv. Mulato II	Gompertz	0.459 *	1.705 *	0.117 *	-	*t* * = 14.55 days	96.47	1.27	<0.05
						P*_t_* = 0.17 g			
*B. brizantha*	Richards	0.553 *	0.823 *	0.125 *	0.257 *	*t* * = 17.45 days	90.10	2.54	<0.05
						P*_t_* = 0.23 g			

* Significant estimate by *t*-test (α = 0.05).

**Table 4 insects-11-00349-t004:** Duration (mean ± SEM) of development stages and pupae weights (mean ± SEM) of *S. frugiperda* in the grasses.

Grass	Duration of Cycle Components (Days)					Complete Cycle (Days)	Pupae Weight (g)
	Egg	Larva	Prepupae	Pupae	Adult		
*Z. mays*	3.25 ± 0.09 B	14.00 ± 0.00 C	1.62 ± 0.18	6.86 ± 0.40	10.43 ± 0.20	36.08 ± 0.60 C	0.125 ± 0.012 A
*P. maximum* cv. Mombasa	3.55 ± 0.11 AB	21.25 ± 0.37 A	2.62 ± 0.18	7.50 ± 0.38	10.25 ± 0.17	45.17 ± 0.53 A	0.119 ± 0.006 AB
*S. sudanense*	3.30 ± 0.10 B	17.22 ± 0.15 B	2.16 ± 0.16	6.83 ± 0.65	11.33 ± 0.33	40.80 ± 0.72 B	0.104 ± 0.007 AB
*P. maximum* cv. Tanzania	3.60 ± 0.11 AB	20.28 ± 0.68 A	2.50 ± 0.34	7.40 ± 0.40	10.80 ± 0.20	45.10 ± 0.86 A	0.115 ± 0.005 AB
*L. multiflorum*	3.30 ± 0.10 B	15.22 ± 0.22 C	1.55 ± 0.24	6.28 ± 0.28	10.86 ± 0.34	36.73 ± 0.29 C	0.099 ± 0.007 AB
*B. brizantha*	3.80 ± 0.09 A	19.57 ± 0.67 A	2.14 ± 0.55	7.43 ± 0.29	10.57 ± 0.29	45.00 ± 0.49 A	0.088 ± 0.007 B
*B. hybrid* cv. Mulato II	3.65 ± 0.11 AB	20.33 ± 0.50 A	2.33 ± 0.29	7.67 ± 0.17	10.78 ± 0.28	45.27 ± 0.46 A	0.095 ± 0.003 AB
F	3.994	45.118	2.007	1.864	1.608	55.559	3.434
*p*	0.001	<0.05	0.084	0.110	0.169	<0.05	0.007

* Means in a column followed by different letters indicate significant differences by Tukey test (α = 0.05).

**Table 5 insects-11-00349-t005:** Component loadings matrix for trap plants selection variables for *S. frugiperda*.

Variables	Principal Components		Communalities
	CP1	CP2	
Larval growth performance	−0.932	−0.090	0.877
Cycle duration	0.919	−0.098	0.854
Oviposition	0.829	0.379	0.831
Attraction	0.723	0.537	0.810
Feeding	−0.545	0.523	0.570
Survival	−0.578	0.712	0.841
Eigenvalues	3.553	1.230	-
Variance (%)	59.21	20.50	-
Cumulative variance (%)	59.21	79.71	-

**Table 6 insects-11-00349-t006:** Classification (in decreasing order of suitability) of trap plants according to the TRAPS index.

Species	TRAPS	Ranking
*Panicum maximum* cv. Mombasa	388.699	1
*Panicum maximum* cv. Tanzania	299.441	2
*Brachiaria hybrid* cv. Mulato II	290.397	3
*Sorghum sudanense*	289.642	4
*Brachiaria brizantha*	278.859	5
*Zea mays*	239.517	6
*Lolium multiflorum*	164.394	7

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
