# Peer review of "Plant Selection for the Establishment of Push–Pull Strategies for Zea mays–Spodoptera frugiperda Pathosystem in Morelos, Mexico"

_insects, 2020, doi:10.3390/insects11060349_

Round 1

Reviewer 1 Report

The authors conducted a series of bioassays aiming at parametrizing a selection index for plants (TRAPS) based on PCA to be used as a tool to select a pool of crop and intercropped plants. Subsequently, the selected plants are proposed as traps and repellents respectively to derive push-pull system (an environmentally friendly managing scheme) able to reduce the damage infringed by the pest Spodoptera frugiperda in maize crops.

I found the manuscript quite interesting and well structured. The writing style allows easily following the whole pipeline of ideas, especially taking into account the high amount of information presented.

Methods are adequately depicted and, in general, statistics follow the proper analysis methods according to the type of data generated in each experiment. Although I miss a kind of multinomial model for the olfactometry assay I especially liked this part and the nice presentation of results. The results found in the study are well discussed and integrated with related literature. In my opinion, the manuscript deserves publication after some minor changes.

Line 78: duplicated “in relation”.

Line 84: Zea -> Z.

Line 91: Missing space at 24

Line 92: “20x20x20” Please use symbol instead of letter “x”. (Same thing hereafter).

Line 105: Do not use italics for species names in subheadings. (Same thing hereafter).

Line: 108: Were the two females considered as a single sample?

Line 109: Extra spaces at “Figure 1 B”.

Line: 110: Were the four females considered as a single sample? If so, I would not call “replicate” each trial, maybe “repetition”.

Line 128-129: Hard to read, please rephrase.

Line 142: Was the airflow somehow filtered?

Line 179: What is ISPT? Did you mean TRAPS?

Line 180: “PCA”.

Line 188: I would delete “(Expression 1)”.

Line 231: “Levene's”.

Line 231: Provide the reference for SPSS

Line 242 and 245: Capitalize 6A and 6B.

Line 274: “B. brizantha”.

Line 299: Does “gl” stands for “grados de libertad”? If so, replace by “df”. (Same thing hereafter).

Line 304: exercised -> exerted?

Line 305-306: This is an interpretation, not a result. Delete or move to discussion.

Line 307: “T. erecta”.

Line 336 and 338: Capitalize 11A and 11B.

Line 352: Capitalize Tukey.

Line 373,374: I would avoid naming a new expression since it is derived from Exp 1.

Lines 377-384: ¿Repetition of the previous paragraph?

Line 392-393: “whose field...studies”. This is not a result. Move to discussion.

Table 4 (First line, last column), Table 5 (First line, last column), Table 6 (Last line first column): Why don’t you use just “P”.

Figure 9B: Please place the name of the plants (x-axis) within the legend.

Reviewer 2 Report

The manuscript by Mariel-Guera et al carried out the plant selection for development of push-pull system against fall armyworm (FAW). This type of study is essential of sustainable plant protection strategy development against FAW. Therefore, this manuscript looks like to be interesting to the potential readers of this journal and merit to publication. But this manuscript is not acceptable in the present form because several shortcomings are found.
Most serious problem is redundancy. I point out several sections but please check and revise all of the text. Second, so authors did not describe the related table/figure, it’s very hard to understand the discussion section but it seems the points are not clear. I need to see revised discussion section before I evaluate this section.
Therefore, my suggestion of this manuscript is Major Revision.

Introduction:
This section is verbose. Especially, the 2nd paragraph (L26 to L53), 4th to 5th (L62 to L78) should be shorter.

Materials and Methods:
This section is also redundant. The text and Figures are duplicate. Figure 1 should be deleted. Regarding the arenas, it’s enough to describe in text because the condition is simple. Figure 3 and 5 are also same. Please consider deleting these figures. In addition, Table 2 should be deleted or should be moved to supplementary material.

Results:
“3.2. Influence of trichome density on oviposition preference (L263 to L274)” can be shorter. And please consider moving Table 3 to supplementary material.
Table 4: “Sig.” means P value?

Discussion:
Hard to find the related data. Authors should be described related table and/or figure etc. Even though several models were tried to fit the data set at Table 5, I do not understand how this analysis results contributed the author’s finding.

Reviewer 3 Report

This manuscript is about push-pull strategy for the control of insect pest.

All experiments were well designed,and results and discussion is well described.

However, some points should be addressed.

Line 133-142: Please include the photo of 4-way olfactometer

Line 133-139: Description is not clear. Did author used 80 larvae for each attractant-repellent plant combination? How many larvae were tested for each replication? How long for each bioassay?

Line 145 3cm, 2cm --> 3 cm, 2 cm

Line 154-161. Did author switch test plants? How long observed for each bioassay?

Figure 10. Describe the number of insect that showed no response.

                Deacribe *: p<0.05 at Figure caption.

Round 2

Reviewer 2 Report

 The manuscript has been revised well.